# A Deep-Ensemble-Learning-Based Approach for Skin Cancer Diagnosis

Khurram Shehzad [1], Tan Zhenhua [1], Shifa Shoukat [2], Adnan Saeed [3], Ijaz Ahmad [4], Shahzad Sarwar Bhatti [5] and Samia Allaoua Chelloug [6,*]

1 Software College, Northeastern University, Shenyang 110169, China; khurram.6470@gmail.com (K.S.); tanzh@mail.neu.edu.cn (T.Z.)
2 National Center for Bioinformatics, Quaid-i-Azam University, Islamabad 15320, Pakistan; shifabinteshoukat@gmail.com
3 Department of Computer Science& Information Technology, Lahore Leads University, Lahore 54990, Pakistan
4 Institute of Computer Science and Information Technology (ICT/IT), Agriculture University Peshawar, Peshawar 25130, Pakistan
5 Department of Information Sciences, Division of Science and Technology, University of Education, Lahore 54000, Pakistan; shahzad.sarwar@ue.edu.pk
6 Department of Information Technology, College of Computer and Information Sciences, Princess Nourah bint Abdulrahman University, Riyadh 11671, Saudi Arabia
* Correspondence: sachelloug@pnu.edu.sa

**Abstract:** Skin cancer is one of the widespread diseases among existing cancer types. More importantly, the detection of lesions in early diagnosis has tremendously attracted researchers' attention. Thus, artificial intelligence (AI)-based techniques have supported the early diagnosis of skin cancer by investigating deep-learning-based convolutional neural networks (CNN). However, the current methods remain challenging in detecting melanoma in dermoscopic images. Therefore, in this paper, we propose an ensemble model that uses the vision of both EfficientNetV2S and Swin-Transformer models to detect the early focal zone of skin cancer. Hence, we considerthat the former architecture leads to greater accuracy, while the latter model has the advantage of recognizing dark parts in an image. We have modified the fifth block of the EfficientNetV2S model and have included the Swin-Transformer model. Our experiments demonstrate that the constructed ensemble model has attained a higher level of accuracy over the individual models and has also decreased the losses as compared to traditional strategies. The proposed model achieved an accuracy score of 99.10%, a sensitivity of 99.27%, and a specificity score of 99.80%.

**Keywords:** skin cancer; dermoscopy; Swin-Transformer; EfficientNetV2S; ensemble model; deep ensemble learning

## 1. Introduction

As per the World Health Organization (WHO), skin cancer ranks among the most prevalent forms of cancer [1]. It represents 33% of all identified types of cancer and it is among the most prominent cancer types during the present decade [2]. The abnormal growth of skin cells is considered the main cause of skin cancer and it is affected by the decrease in the ozone layer, which protects people against ultraviolet (UV) radiation [3]. Squamous cell carcinoma (SCC), actinic keratosis (solar keratosis), melanoma (Mel), and basal cell carcinoma (BCC) are the major types of skin cancer [4]. Seventy-five percent of fatalities related to skin cancer are attributed to melanoma, which constitutes the most dangerous form of the disease [5]. People with fair skin, a history of sunburns, those who overexpose themselves to UV light, and those who use tanning beds are more likely to be affected by skin cancer [6].

The importance of early and precise identification in the treatment of skin cancer cannot be overstated. If melanoma is not detected early, it grows and spreads throughout

the outer layer of the skin, eventually penetrating the deeper layers and connecting with the blood and lymph vessels. Therefore, it is crucial to detect it early to provide appropriate medication for the patient. The range of the anticipated five years survival rate for patients with diagnoses is 15% if caught late to over 97% if caught early [7]. Early detection is therefore essential for skin cancer treatment [8]. Oncologists frequently utilize the biopsy method to diagnose skin cancer. To verify if a suspicious skin lesion is malignant or not, a sample must be taken. However, considerable effort and time are required for the diagnosis.

It is worth mentioning that computer-assisted identification of skin cancer symptoms is more convenient, inexpensive, and faster. To study the symptoms of skin cancer and identify whether they are caused by melanoma or not, numerous noninvasive methods are available such as dataset access, preprocessing of datasets, application of segmentation after data preprocessing, necessary feature extraction, and classification after the diagnosis process. Many machine-learning techniques exist for the detection of various kinds of cancer. In particular, CNNs are frequently employed by researchers to identify skin cancer and for classification of skin lesions [9].

CNN models have significantly outperformed highly qualified healthcare practitioners in the categorization of skin malignancies. Early skin cancer classification derives characteristics from skin cancer images using manual feature extraction methods such as shape, texture, geometry, and other factors [10]. Currently, with the emergence of deep artificial intelligence (DAI), the technology has advanced significantly in learning the study of imaging in medicine. In the area of identifying medical images, the categorization of skin cancer successfully uses CNN, which is widely used and has high accuracy [11]. With the advancements in the field of technology, researchers in the field of AI have designed some innovative techniques for the detection of skin cancer with greater accuracy. Among the latest models, the Shifted Windows (Swin) deep-learning-based model [12] is the improved form of the Vision Transformer (ViT) [13] model, which has shown efficient performance with higher accuracy; that is why this model, which generated improved results, was selected.

Motivated by the synergy of computer vision and natural language processing (NLP), it is advantageous to use both disciplines since it makes it easier to model visual and textual signals together and allows for deeper modeling knowledge sharing. The Swin-Transformer model has impressive results across a variety of vision issues that deepen the mindset in the community and support integrating the modeling of linguistic and visual signals. In the presented work, we developed the deep ensemble learning model that uses both EfficientNetV2 and Swin-transformer models for the classification of the multi-class skin diseases dataset. We have modified the fifth block of EfficientNetV2 and integrated the Swin-Transformer model. Then, we merged the outputs of the two modified models. The results of the proposed strategy have been validated through experiments on a well-known dataset, namely the HAM-10000 dataset, in terms of improved accuracy score, sensitivity, and specificity score. In summary, we have performed the following contributions:

- We have designed a novel deep ensemble learning approach to classify multi-skin disease datasets.
- To improve the efficiency, we combined both the EfficientNetV2S and Swin-Transformer models.
- To improve the image disease area, we applied multiple data preprocessing techniques.
- We applied a data augmentation technique to balance the skin images which speeds up the training process of the proposed model.
- We validated the results of our model on a famous dataset, namely HAM-10000.
- The experimentations show increased performance in terms of accuracy score, sensitivity, F1-score, and specificity score.

The subsequent sections of this paper are arranged as follows. In Section 2 (Related Work), an overview of diverse models and outcomes pertaining to the diagnosis and classification of skin cancer is presented. Section 3 details the research methodology employed in this study, along with a thorough discussion of the approach. Subsequently,

Section 4 presents an evaluation of the outcomes and a comprehensive discussion thereof. Finally, in Section 5, a succinct conclusion is provided along with a brief outline of the future research prospects.

## 2. Related Work

In this section, we discuss the existing work in the field of skin cancer detection to obtain a deeper understanding of deep learning models and their working with findings.

With a small amount of training data, Yuet al. [14] built a highly deep CNN and a variety of learning frameworks. Esteva et al. [15] employed a pre-trained CNN approach to develop and acquire a dermatologist-level diagnosis from more than 120,000 images. Haenssle et al. [16] presented CNN models that have proven superior to or more reliable than dermatologists. Deep learning is used to create further techniques, such as the ensemble model [17], which aggregates the features of numerous models, to identify skin cancer.

Esteva et al. [18] conducted research on the application of a pre-trained Google Inception-V3CNN model to improve the categorization of skin cancer. The study employed a dataset of 129,450 clinical skin cancer images, out of which 3374 were dermoscopic images. The results reported an accuracy of 72.1 ± 0.9 in skin cancer categorization using the aforementioned model, which was evaluated on the ISCI 2016 challenge dataset [19].A CNN with over 50 layers was built in 2016 for the categorization of malignant Mel skin cancer. In competition, the challenge's highest categorization accuracy was 85.5 percent. In [15], the researchers applied a deep CNN for the classification of clinical images related to 12 types of skin abnormalities. Their results showed an accuracy of 96%. This research did not focus on a detailed review of the classifier. The authors presented a detailed systematic review of the deep learning (DL) classifiers in [14].

In another study [20], the researchers suggested a deep convolutional neural network (DCNN)-based categorization of skin lesions and an optimized color feature (OCF) for lesion segmentation. A hybrid strategy is used to get rid of the artifacts and boost lesion contrast. Then, a color segmentation approach called OCFs was introduced. An already-existing saliency strategy that is combined with a brand-new pixel-based method dramatically improved the OCF approach. The proposed models achieved 92.1%, 96.5%, and 85.1% on each of the three datasets, which demonstrates the provided method's excellent performance. In [21], a ResNet model that was previously trained on non-medical datasets was fine-tuned using a small quantity of a combined dataset from three distinct sites. The results demonstrated a 20-point improvement in performance on melanoma (Mela), malignant (skin cancer), and benign mole detection using the previous knowledge learned from photographs of mundane objects from the Image-Net collection. According to the results, skin moles can be classified using features from non-medical data, and the distribution of the data has an impact on how well the model performs.

In [22], the researchers proposed a machine-learning approach for diagnosing dermatological illnesses using images of lesions as opposed to the traditional method, which relies on medical professionals. The proposed model was created in three stages, the first of which involved committing to data collection and augmentation, the second of which involved model development, and the third of which concerned prediction. In this work, they used image processing technologies with a variety of AI algorithms, such as the ANN, to create a better structure and achieved 89% accuracy.

In their study, the authors of [23] proposed a novel technique for the classification of skin lesions that incorporates the use of deep learning features fusion and machine learning features. The proposed approach consists of a five-step methodology that involves the following: contrast amplification, image acquisition, DL feature extraction, feature selection utilizing a hybrid method that combines WO, and EMI, and the integration of selected features using a modified canonical correlation-based technique, which is subsequently followed by extreme-learning-machine-based classification. The feature selection process has been shown to enhance the precision and computation speed of the system. The

experimental evaluation of the proposed method was performed using HAM-10000 and ISIC-2018, two publicly accessible datasets, achieving an accuracy of 93.40% and 94.36%, respectively, on both datasets.

In the research study of [24], a hybrid technique was proposed to integrate the binary images produced by the intended 16-layered convolutional neural network (CNN), resulting in enhanced contrast in high-dimensional data and network model Saliency segmentation using the high-dimensional cosine transform (HDCT) to make use of a maximum amount of information recovered from the binary picture. The maximal mutual information approach was presented which returned the RGB lesion image with segments. A pre-trained DenseNet201 model was used in the classification module, which was re-trained via using transfer on segmented lesion images. Another study [25] classified the Human Again Machine (HAM-10000) dataset into seven skin cancer categories. The researchers developed a two-tier architecture model, and in the first phase they applied different data augmentation approaches to enlarge the dataset in the second tier; the researcher applied Medical Vision Transformer for the data analysis and gained better results from the previously carried out research. However, the proposed model differs from the existing methods.

In most of the proposed methods, the researchers applied a single CNN model for the detection of skin cancer. We designed a hybrid approach for skin cancer detection. Some researchers utilized only color normalization to enhance the dataset's quality, while others relied on image segmentation strategies to obtain higher accuracy. However, we applied a comprehensive image processing strategy to enhance the quality of the dataset. The main objective of this study is to design a deep ensemble learning model for the diagnosis of skin cancer with groundbreaking results. We applied EfficientNetV2S and Swin-Transformer models to build a deep ensemble model for skin cancer diagnosis which benefits from the diversity of the individual models while still maintaining interpretability. This approach affirmed the novelty of this research work. Previous research work has mainly focused on diagnosing two types of skin cancer, i.e., melanoma and non-melanoma, but we classified seven categories of skin cancer in this proposed work.

## 3. Methods and Materials

This section provides the overall procedure of our model and the main building block of our paper.

Figure 1 represents the complete system architecture of the proposed methodology of the research work.

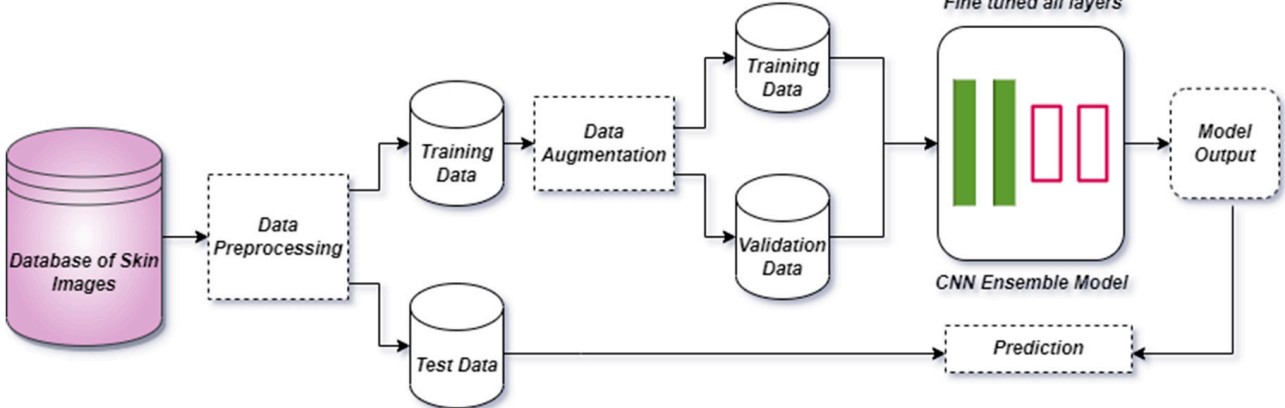

**Figure 1.** System architecture of the proposed methodology.

### 3.1. Dataset Description

In this manuscript, we employed the HAM-10000 image dataset, which was created by the International Skin Image Collaboration (ISIC) in 2018 [26] and is publicly accessible. The HAM-10000 dataset comprises seven types of dermoscopic images of skin cancer,

which were sourced from diverse populations and acquired by various modalities. The selected dataset includes multiple images of a similar lesion, collected at different times and under different lighting conditions, which provides additional variability to the dataset. This can help to improve the robustness of the ensemble learning model by enabling it to handle variations in image quality and illumination. The HAM-10000 dataset was primarily compiled by two organizations from two countries: Cliff Rosendahl's organization from Queensland, Australia, and the Dermatology Department of the Medical University of Vienna, Austria. The assemblage of this dataset was the culmination of two decades of tireless effort [27]. The weight of each of the seven categories is depicted in Figure 2.

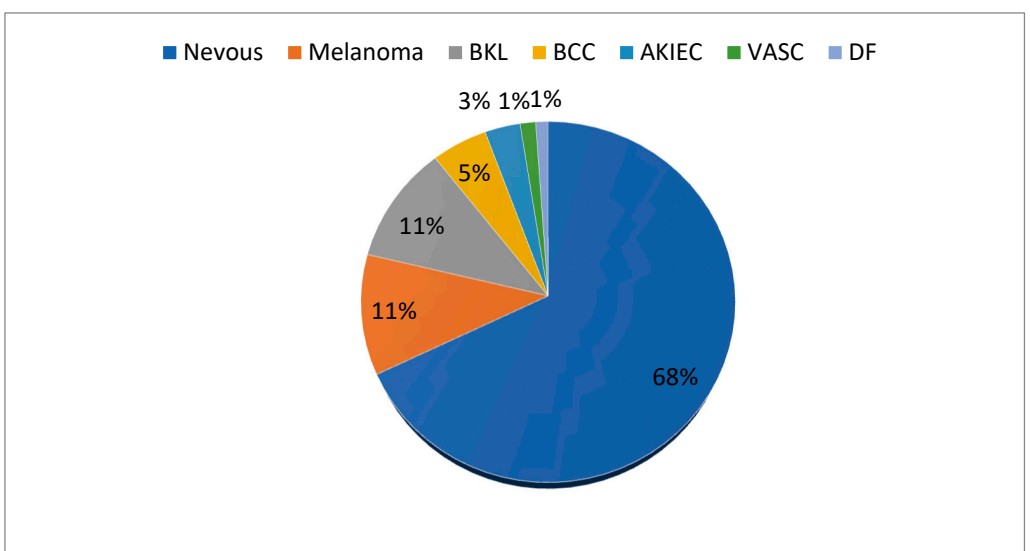

**Figure 2. The** HAM-10000 dataset divided into seven categories by percentage.

*3.2. Data Preprocessing*

For multi-class skin disease problems, we applied different image preprocessing techniques on the image dataset to improve the image quality and disease area. Firstly, we applied the color normalization technique by using the gray world algorithm. The latter is the simplest technique that allows the red, green, and blue channels' average intensities to be equal. If P (I,j) is the original RGB image with a size of r × s and R (I,j), B (I,j), and G (I,j) are the channels in the image, then the green channel is typically kept untouched and red and blue channels gains are calculated as follows:

$$R_{gain} = \frac{\mu_G}{\mu_B} \tag{1}$$

And

$$B_{gain} = \frac{\mu_G}{\mu_R} \tag{2}$$

where $\mu_R$, $\mu_B$, and $\mu_G$ are the channel's respective average intensity levels.

Some of the images in the skin cancer dataset have hidden regions due to skin hair. Consequently, prior to further analysis or processing, it is necessary to perform preprocessing of the skin images with the objective of removing artifacts that are present in the images. For this purpose, we applied morphological filtering to clean the presence of hair from the skin images. We also applied the crop function to eliminate the unnecessary part of the image. All steps of image preprocessing are shown in Figure 3. After that, we reduced the image size from 450 × 600 to 384 × 384 resolutions.

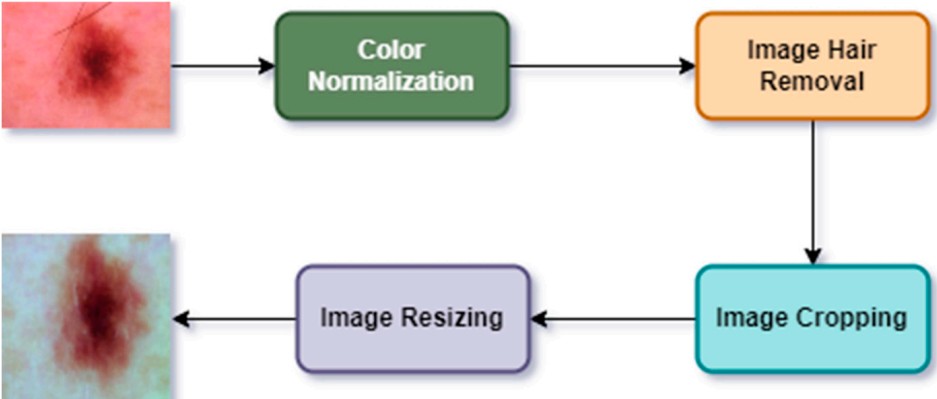

**Figure 3.** Image preprocessing steps.

### 3.3. Data Augmentation

Data augmentation is a widely adopted technique in conjunction with convolutional neural network models, particularly in scenarios where the image dataset is imbalanced. In this context, we noted that HAM-10000 is a multi-class skin dataset that is modest in size and also exhibits imbalanced class distribution. Specifically, certain classes possess fewer images compared to others, creating an imbalance in the dataset. The application of data augmentation methods effectively addresses both the size and imbalance disparities, as depicted in Figure 4. It is noteworthy that all image categories carry equal weight, accounting for 14.3% of the dataset.

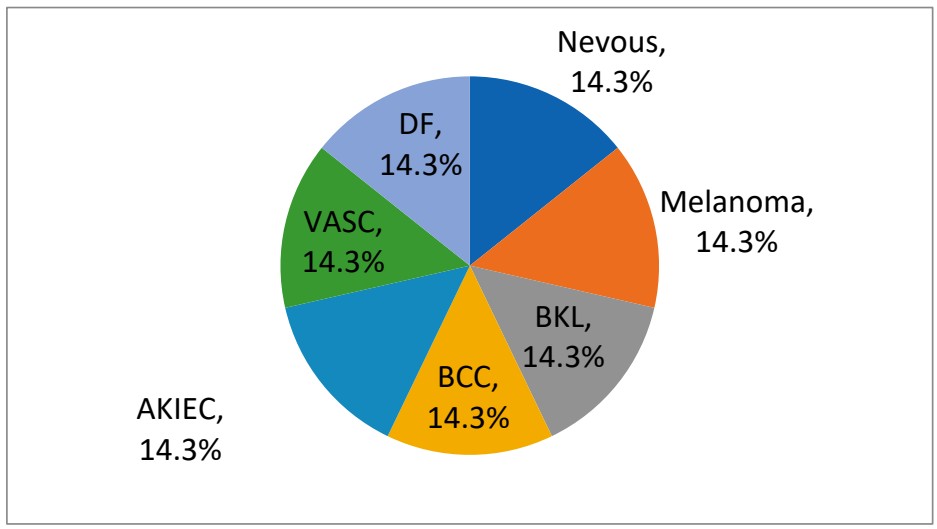

**Figure 4.** The augmentation dataset in its balanced form.

Various augmentation techniques were employed to achieve a balanced distribution of the skin disease dataset, as depicted in Figure 5. Specifically, the images were randomly rotated up to 30 degrees and zoomed by 10%. Additionally, both horizontal and vertical flips were incorporated. A shear range, height shift range, and width shift range of 10% were applied randomly to the images. To modify the image's color, a brightness range of [0.5, 1.2] was applied. Following the augmentation process, the multi-class skin disease dataset comprised 42,238 images, and all classes exhibited a balanced distribution.

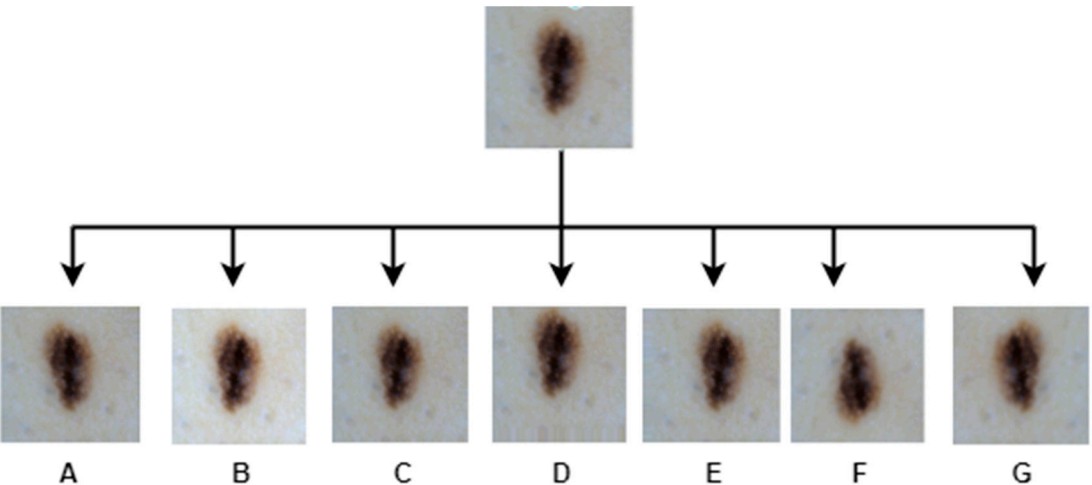

**Figure 5.** Real image (at the **top**) and augmented images (**A–G**).

### 3.4. Transfer Learning

It is a common practice of using CNN models to train new data by using previous knowledge. Transfer learning can save a significant amount of time and resources, especially for complex models that require extensive training. Transfer learning can also improve the accuracy of new models by leveraging the acquisition of knowledge from a pre-existing trained model has the potential to yield superior results within a limited timeframe. As such, we have employed a fine-tuning methodology based on transfer learning, whereby all the layers of our models have been trained.

### 3.5. The Swin-Transformer Model

The Swin-Transformer model is built upon two fundamental concepts, namely the-hierarchical feature map (HFM) and shifted window attention (SWA), which have been developed to address the major challenges faced by the original vision transformer (ViT). The Swin-Transformer architecture comprises two core blocks, namely the Patch Merging Block (PMB) and Swin-Transformer Block (STB). HFMs are intermediate tensors generated hierarchically that facilitate down-sampling from one layer to the next. The Swin-Transformer employs a convolution-free down-sampling technique referred to as patch merging, where a feature map's smallest element is represented as a "patch". A $14 \times 14$ feature map contains 196 patches. Patch merging combines the features of each set of $n \times n$ adjacent patches and assigns them to the down-sampled features, resulting in a reduction in size by a factor of $n$. The Swin-Transformer framework employs alternative modules called Window MSA (W-MSA) and Shifted Window MSA (SW-MSA) modules, in place of the conventional multi-head self-attention (MSA) modules utilized in the ViT architecture. The first sub-unit integrates the Window MSA (W-MSA) module, while the second sub-unit incorporates the Shifted Window MSA (W-MSA) module.

### 3.6. EfficientNetV2S

EfficentNetV2S is a popular training model recognized for its fast training speed, efficient parameter utilization, and compact size, which is approximately 6.8 times smaller than other models. With its training approach, EfficientNetV2S demonstrates a strong capability to process the Image-Net dataset [28]. The depth-wise convolution layer, which is used by EfficientNetV2S, requires fewer parameters and float-point operations per second (FLOPS) but is unable to fully utilize contemporary accelerators such as GPU/CPU.EfficientNetV2S is pre-trained on the large-scale Image-Net dataset, which helps to improve its performance in a variety of computer vision tasks [29]. EfficientNetV2S is designed to be optimized for resource-constrained devices, with small model sizes and a low memory footprint. This makes it well-suited for deployment on mobile phones and embedded systems.

EfficientNetV2S utilizes the BMConv layer in combination with the fused-MBConv layers in its initial layer to enhance computational efficiency. In recognition of the reduced memory access cost associated with smaller expansion ratios, this model implements a reduced expansion ratio for BMConv. Additionally, EffiecientNetV2S adopts 3 × 3 kernel sizes that are comparatively smaller and compensate for the smaller receptive field by incorporating a greater number of layers.

*3.7. Ensemble Strategy*

EffcientNetV2S was employed as the foundational model, with an input size of 384 for the image. The fifth block of the base model was subsequently adjusted, and the block_5_02_expand activation layer was utilized as input for the Swin-Transformer model. We obtained 24 output sizes from the combined output of the adapted model and base model, and this merged layer was then processed by the final classification layer.

Furthermore, our pipelines utilize the technique of gradient accumulation for model construction. Transformer-based models are known to be computationally intensive, and as a result, the batch size may be restricted. To address this issue, we opt to partition the batch into smaller mini-batches, which are subsequently executed in a sequential manner while their outcomes are aggregated. As a consequence, this approach can overcome memory constraints and enable the model to be trained using less memory than it would with a larger batch size. With execution gradient accumulation with 8 steps and a batch size of 8 images, the proposed model achieved comparable results to a batch size of 64 images. Figure 6 illustrates the operation of the deep ensemble model.

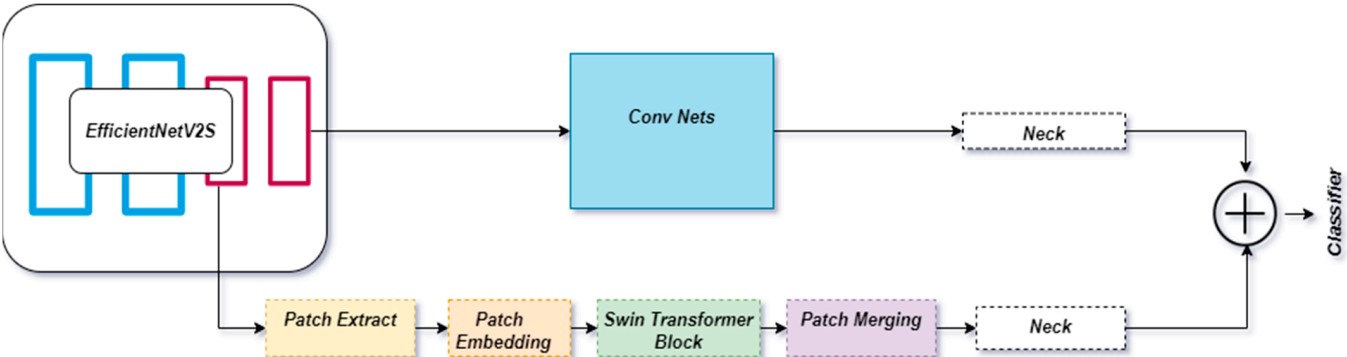

**Figure 6.** Workings of the proposed ensemble model.

## 4. Results and Discussion:

*4.1. Evaluation Methods*

The utilization of a confusion matrix is a valuable approach to assess model performance, particularly for multi-class problems. The confusion matrix provides a comprehensive representation of the correctly classified *TP* values, *FP* values that are categorized in the wrong class, false negative (*FN*) values that belong to the incorrect class, and correctly classified *TN* values in the other class [30]. To measure the effectiveness of a model, several commonly used performance metrics are calculated from the confusion matrix, including *ACC*, *P*, *Sn*, *Sp*, and F-score [31–33]. These metrics are derived from equations formed using the confusion matrix, enabling an accurate assessment of the proposed model's efficiency.

$$ACC = \frac{TP + TN}{TP + FP + TN + FN} \tag{3}$$

$$LOSS = 1 - ACC \tag{4}$$

$$P = \frac{TP}{TP + FP} \tag{5}$$

$$Sn = \frac{TP}{TP + FN} \tag{6}$$

$$Sp = \frac{TN}{TN + FP} \tag{7}$$

$$F1 = 2\frac{(P \times Sn)}{(P + Sn)} \tag{8}$$

$$TPR = \frac{TP}{TP + FN} \tag{9}$$

$$TNR = \frac{FP}{FP + TN} \tag{10}$$

### 4.2. Experimental Settings

In addressing the multi-class skin disease problem, a systematic approach was taken to partition the image data into distinct subsets of training, testing, and validation. Specifically, the test data were comprised of 1002 images, representing 10% of the original dataset, and were subject to rigorous data processing steps. The training and validationsets, containing 3502 and 6336 images, respectively, were also subjected to similar processing procedures, including the incorporation of augmented images.

### 4.3. Model Training and Testing

The main purpose of model training and testing is the minimization of error and to improve the overall accuracy of the proposed technique. To understand the error function in the research work, Equation (3) is given below:

$$E(w) = \frac{1}{K \times N} \sum_{K=1}^{K} \sum_{n=1}^{N_L} \left( Y_n^k - d_n^k \right)^2 \tag{11}$$

Equation (11) $Y_n^k$ represents the actual output images in the model and $K$ denotes the input images and desired output vectors. $X^k$ is the $K$th-trained image and $d^k$ is the desired output vector. With the help of error sensitivities, we measure the error gradient, which is equal to the partial derivatives of the error function.

We set the learning rate to 0.001 for both models with the Adamax optimizer. The batch size and epochs were set to 16 and 20, respectively, for training. We set a patience value of 1 and a stop patience value of 3. For calculating loss, a categorical cross-entropy function was applied. The model saves the best results for the validation set. All of the experiments were implemented in the Python 3.7 version and TensorFlow platforms. The next section includes the experimental part of both models.

### 4.4. EfficientNetV2S vs. Ensemble Model

We applied CNN EfficientNetV2S and ensemble (EfficientNetV2S + Swin-Transformer) models to achieve better performance. In this section, we compare the different parameters of both models and describe the importance of the proposed model for the classification of a multi-class skin disease problem.

Table 1 provides a comprehensive overview of the training procedure for both models, including details on every epoch's training and validation values. The EfficientNetV2S model is trained for 18 epochs, while the ensemble model is trained for 13 epochs. Our analysis revealed that the ensemble model learns more efficiently and requires less time than the single EfficientNetV2S model. While the accuracy of both models is similar, as depicted in Table 1, the proposed ensemble model outperforms the EfficientNetV2S model in terms of training and validation loss, demonstrating its superior ability to identify diseased areas in images. Figure 7 provides a visual representation of the accuracy and validation curves of both models. The ensemble model's training is stopped after 13 epochs

since the validation loss stopped improving after 10 epochs. Similarly, the EfficientNetV2S model is stopped after 18 epochs. Both models show no signs of over-fitting as the training loss decreases from the validation loss during the training procedure.

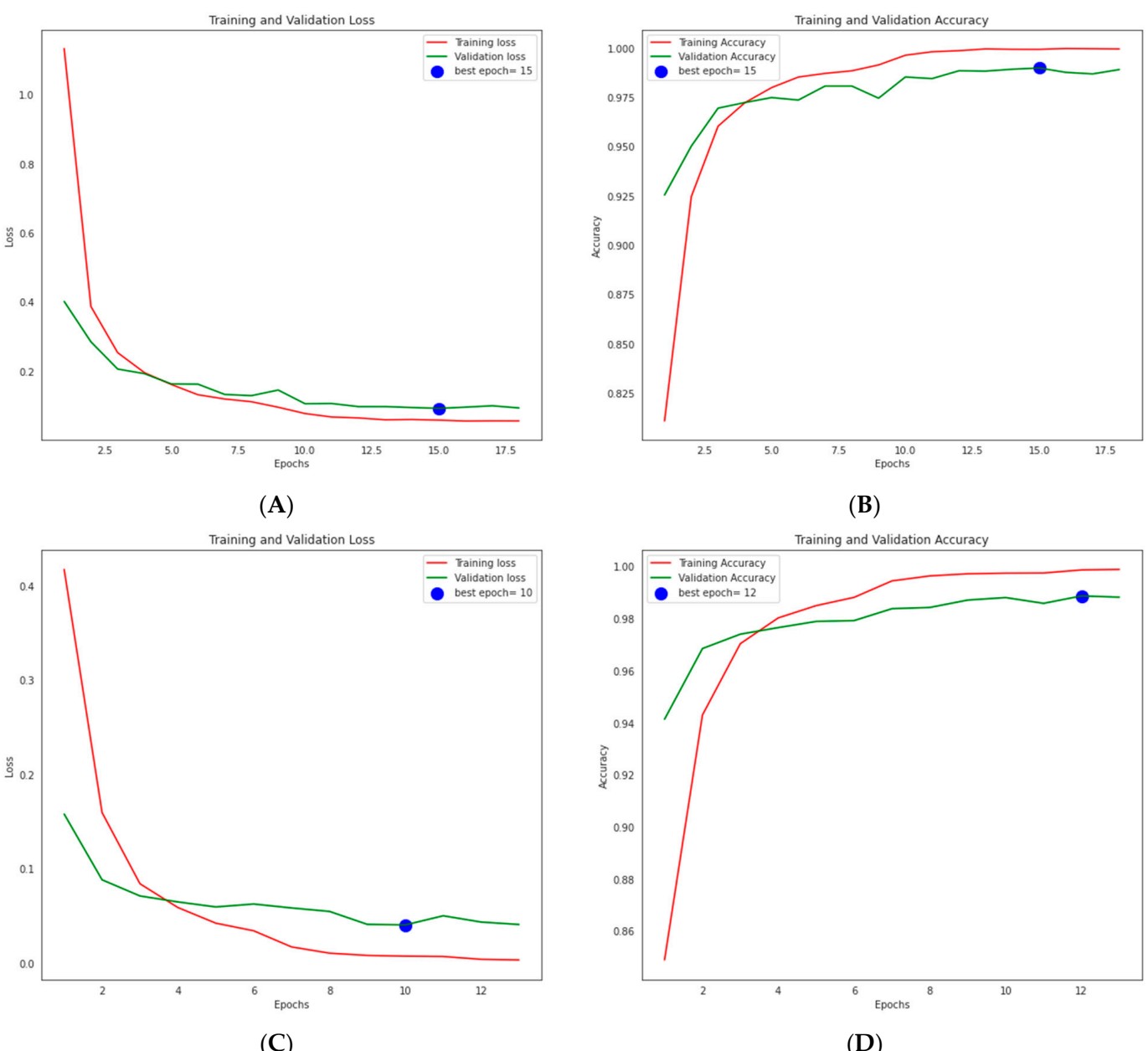

**Figure 7.** Training and validation accuracy and loss of (**A**,**B**) the EfficientNetV2S model and (**C**,**D**) the ensemble model.

**Table 1.** Full training procedure of both models with training and validation values.

| | EfficientNetV2S | | | | | Ensemble Method | | | |
|---|---|---|---|---|---|---|---|---|---|
| Epochs | Training Accuracy | Training Loss | Valid Accuracy | Valid Loss | Epochs | Training Accuracy | Training Loss | Valid Accuracy | Valid Loss |
| 1 | 81.096 | 1.135 | 92.551 | 0.402 | 1 | 84.903 | 0.418 | 94.145 | 0.158 |
| 2 | 92.468 | 0.388 | 95.028 | 0.285 | 2 | 94.307 | 0.160 | 96.859 | 0.088 |
| 3 | 96.039 | 0.254 | 96.954 | 0.206 | 3 | 97.048 | 0.085 | 97.412 | 0.071 |
| 4 | 97.229 | 0.197 | 97.238 | 0.193 | 4 | 98.036 | 0.059 | 97.664 | 0.065 |

**Table 1.** *Cont.*

| | EfficientNetV2S | | | | | Ensemble Method | | | |
|---|---|---|---|---|---|---|---|---|---|
| Epochs | Training Accuracy | Training Loss | Valid Accuracy | Valid Loss | Epochs | Training Accuracy | Training Loss | Valid Accuracy | Valid Loss |
| **5** | 97.995 | 0.162 | 97.491 | 0.163 | 5 | 98.507 | 0.043 | 97.901 | 0.060 |
| **6** | 98.529 | 0.133 | 97.364 | 0.163 | 6 | 98.825 | 0.035 | 97.932 | 0.063 |
| **7** | 98.716 | 0.120 | 98.074 | 0.133 | 7 | 99.457 | 0.018 | 98.390 | 0.058 |
| **8** | 98.847 | 0.112 | 98.074 | 0.129 | 8 | 99.649 | 0.011 | 98.438 | 0.055 |
| **9** | 99.145 | 0.096 | 97.459 | 0.146 | 9 | 99.730 | 0.009 | 98.722 | 0.041 |
| **10** | 99.635 | 0.078 | 98.532 | 0.106 | 10 | 99.755 | 0.008 | 98.816 | 0.041 |
| **11** | 99.813 | 0.068 | 98.453 | 0.107 | 11 | 99.760 | 0.008 | 98.595 | 0.050 |
| **12** | 99.866 | 0.065 | 98.848 | 0.098 | 12 | 99.877 | 0.005 | 98.879 | 0.044 |
| **13** | 99.955 | 0.060 | 98.832 | 0.098 | 13 | 99.900 | 0.004 | 98.832 | 0.041 |
| **14** | 99.936 | 0.061 | 98.927 | 0.095 | | | | | |
| **15** | 99.933 | 0.059 | 98.990 | 0.093 | | | | | |
| **16** | 99.978 | 0.057 | 98.769 | 0.096 | | | | | |
| **17** | 99.967 | 0.057 | 98.690 | 0.100 | | | | | |
| **18** | 99.953 | 0.057 | 98.911 | 0.094 | | | | | |

### 4.5. Confusion Matrix Analysis

We defined different confusion matrix equations in the evaluation method and the results are shown in Table 2. The proposed strategy of the ensemble model achieves 99.10% accuracy on the test set which is 0.40% higher than the baseline EfficientNetV2S. If we compare other parameters then the ensemble model achieved 99.27%, and 99.80% sensitivity and specificity scores, respectively, which were also 0.19% and 0.35% higher than the baseline EfficientNetV2S scores, respectively. The main progress of the ensemble model has decreased the loss heavily to single efficientNetV2S which shows the better capability of our ensemble model and also evidence that the strategy of a combination of two models works well contrary to the traditional strategy.

**Table 2.** Testing score of the EfficientNetV2S model and the ensemble model.

| Classes | EfficientNetV2S | | | | | | | Ensemble Model | | | | | | |
|---|---|---|---|---|---|---|---|---|---|---|---|---|---|---|
| | Accuracy % | Precision % | Sensitivity % | Specificity % | F1-Score % | Loss | No. of Samples | Accuracy % | Precision % | Sensitivity % | Specificity % | F1-Score % | Loss | No. of Samples |
| AKIEC | - | 1.00 | 1.00 | 1.00 | 1.00 | - | 33 | - | 1.00 | 1.00 | 1.00 | 1.00 | - | 33 |
| BCC | - | 1.00 | 1.00 | 1.00 | 1.00 | - | 51 | - | 1.00 | 1.00 | 1.00 | 1.00 | - | 51 |
| BKL | - | 97.30 | 98.18 | 99.21 | 97.74 | - | 110 | - | 1.00 | 98.18 | 1.00 | 99.08 | - | 110 |
| DF | - | 1.00 | 1.00 | 1.00 | 1.00 | - | 12 | - | 1.00 | 1.00 | 1.00 | 1.00 | - | 12 |
| MEL | - | 96.40 | 96.40 | 99.10 | 96.40 | - | 111 | - | 96.43 | 97.30 | 1.00 | 96.86 | - | 111 |
| NV | - | 99.10 | 98.96 | 97.89 | 99.03 | - | 671 | - | 99.26 | 99.40 | 98.78 | 99.33 | - | 671 |
| VASC | - | 1.00 | 1.00 | 1.00 | 1.00 | - | 14 | - | 1.00 | 1.00 | 1.00 | 1.00 | - | 14 |
| Average | 98.70 | 98.97 | 99.08 | 99.45 | 99.02 | 0.108 | - | 99.10 | 99.38 | 99.27 | 99.80 | 99.32 | 0.037 | - |

Figure 8 shows the confusion matrices of both models. According to Figure 8, the ensemble model only misclassified 9 samples and the effcientNetV2S model misclassified 13 samples, which proved the classification capability of our proposed technique. According to the confusion matrix ensemble model, only 2,3, and 4 images are misclassified as BKL, melanoma, and Nevus, respectively.

### 4.6. ROC–AUC Curve Analysis

The relationship between the sensitivity and FP rate for different threshold settings is known as the receiver operating characteristics (ROC) curve. The ability of a classification model is tested using the area under the curve (AUC), which is the area under this ROC curve. The performance improves with area size. The ensemble model achieved a higher ROC–AUC score than EfficientNetV2S for multi-class skin disease problems as shown in Figure 9.

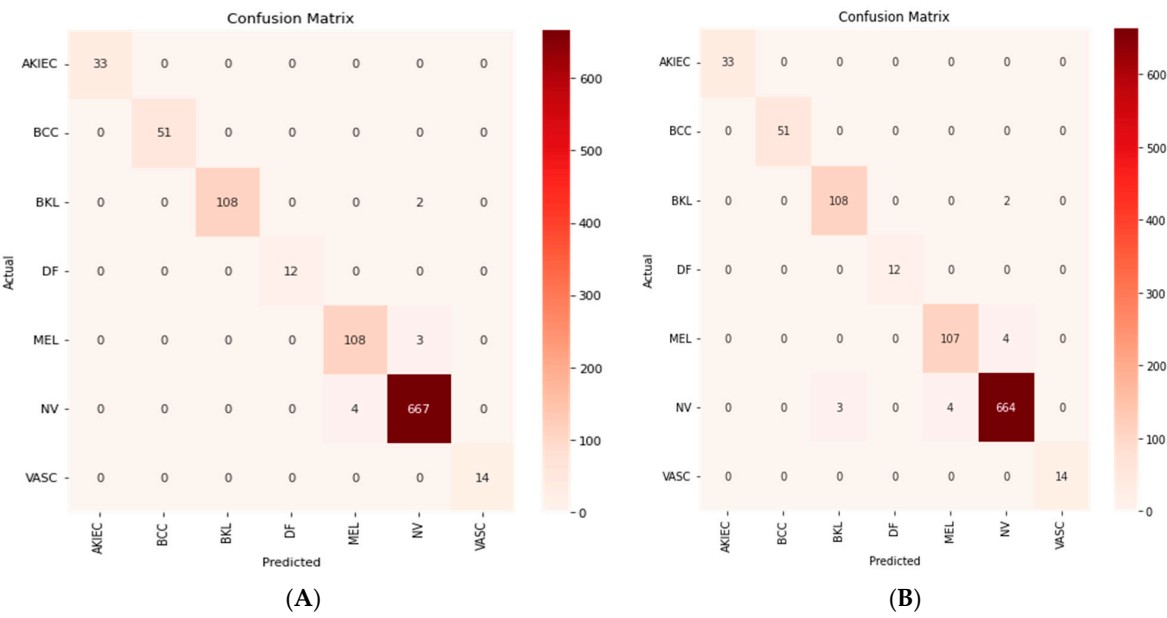

**Figure 8.** Confusion matrix of the proposed ensemble model (**A**) and the baseline EfficientNetV2S model (**B**).

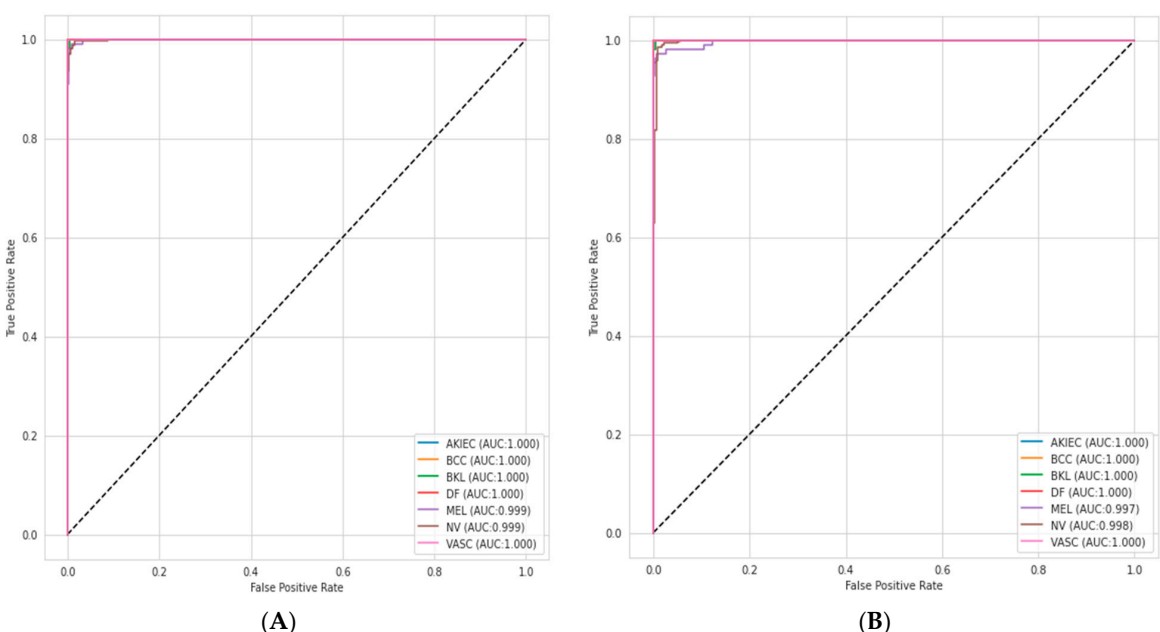

**Figure 9.** ROC–AUC scores of (**A**) the ensemble model and (**B**) the EfficientNetV2S model.

In Table 3, we compared the results of accuracy, specificity, and sensitivity with other research models with the HAM-10000 dataset from the year 2020 to 2022. The results of this ensemble model are at the highest level with regards to accuracy, specificity, and sensitivity.

**Table 3.** Comparison of accuracy, specificity, and sensitivity with other research work using the HAM-10000 dataset.

| Ref. | Year | Research Method | Accuracy (%) | Specificity (%) | Sensitivity (%) |
|------|------|-----------------|--------------|-----------------|-----------------|
| [34] | 2020 | CNN + OVA (one-versus-all) | 92.90 | | |
| [35] | 2021 | MobileNet+LSTM | 85.34 | 92.0 | 88.24 |

**Table 3.** *Cont.*

| Ref. | Year | Research Method | Accuracy (%) | Specificity (%) | Sensitivity (%) |
|---|---|---|---|---|---|
| [36] | 2022 | Modified MobileNetV2 | 91.86 | 92.66 | 91.09 |
| [37] | 2022 | DenseNet169-two classes | 91.10 | 95.66 | 82.49 |
| [38] | 2022 | Cascaded ensemble DL Model | 99.00 | 98.0 | 98.0 |
| **This Paper** | 2022 | EfficientNet-V2 + Swin-Transformer (Ensemble DL Model) | 99.10 | 99.80 | 99.27 |

### 4.7. Grad-Cam Analysis

Grad-Cam is a well-known visualization technique from which we know that the convolutional layers of the models focus on which part of the image. In this paper, the main aim of the Grad-Cam technique is to compare the baseline EfficientNetV2Smodel with the ensemble model. For this purpose, a visualization technique is applied to test the images. After the results, it is clearly shown in Figure 10 that the proposed hybrid model more actively focuses on the target part of the images rather than the baseline EfficientNetV2S and proved that transformer blocks have a greater capacity to see its decision-making process. Additionally, we note that CNN's mid-level feature maps have the propensity to correct more broadly by the transformer blocks.

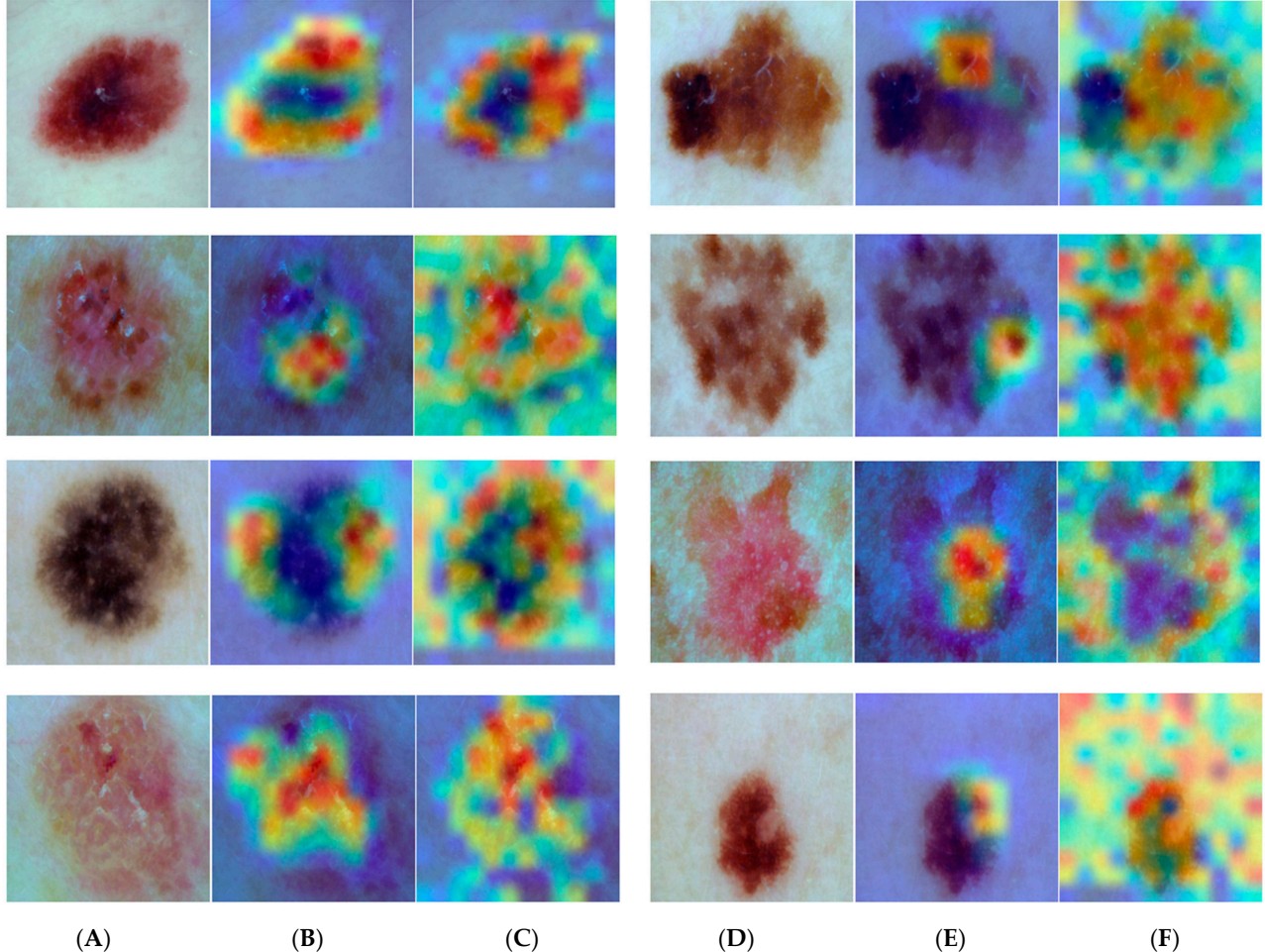

**Figure 10.** (**A–D**) are the real images (**B–E**) visualization of our baseline EfficientNetV2S model (**C–F**) visualization of our ensemble model (EficientNetV2S + Swin-Transformer model).

## 5. Conclusions and Future Work

This paper has presented a deep ensemble learning approach for skin cancer diagnosis that uses the vision of EfficientNetV2S and the Swin-Transformer model to classify multi-class skin disease images. With the help of modification in the fifth block of EfficientNetV2S and with the addition of the Swin-Transformer model, we merged the outputs of both the modified model and the base model. After experiments, the proposed approach achieved higher accuracy rather than the individual models and also heavily decreased the loss which proves that the proposed strategy is more suitable than the traditional strategies. We used HAM-10000, an image dataset that consists of seven types of skin cancer images. We included the data preprocessing techniques to improve the image quality and remove the unwanted part of the image. A data augmentation technique was also utilized to balance the skin image dataset which is necessary for training and to save the proposed model from being biased. The proposed deep ensemble model achieved a 99.10% accuracy score on the test set and, similarly, achieved a 99.27% sensitivity score and a 99.80% specificity score.

Future work for this study could involve the integration of other modalities, such as macroscopic imaging, and clinical information to further improve the accuracy of the diagnostic system. In addition, the development of a mobile application that can assist dermatologists in diagnosing skin cancer using this approach can also be explored. Further studies could investigate the generalization of the approach to different skin types and ethnicities to ensure the accuracy of the approach in a diverse population.

**Author Contributions:** Conceptualization, K.S. and T.Z.; methodology, K.S., S.S. and A.S.; software, S.S. and S.S.B.; validation, K.S., S.S. and A.S.; formal analysis, I.A. and S.S.; investigation, S.S.B.; resources, K.S.; data curation, K.S.; writing—original draft preparation, K.S.; writing—review and editing, K.S. and S.S.B.; visualization, A.S. and I.A.; supervision, T.Z.; project administration, T.Z.; funding acquisition, S.A.C. All authors have read and agreed to the published version of the manuscript.

**Funding:** This work was supported by Princess Nourah bint Abdulrahman University Researchers Supporting Project number (PNURSP2023R239), Princess Nourah bint Abdulrahman University, Riyadh, Saudi Arabia.

**Conflicts of Interest:** The authors declare no conflict of interest.

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
