# Peer review of "A Deep-Ensemble-Learning-Based Approach for Skin Cancer Diagnosis"

_electronics, doi:10.3390/electronics12061342_

Round 1

Reviewer 1 Report

1. A language check is required

2. There are no references to some figures in the text (some are incorrectly numbered)

3.  In Fig. 11, however, it looks as if the base-line model is better focused on the target part (column B and E).

4. There's no need to present the idea of the confusion matrix (Fig. 7)

Reviewer 2 Report

The topic of this work is interesting, but the current version needs major corrections to be acceptable for publishing. 

The corrections are as follows:

1-  The main contributions of this work are not clearly explained. 

2- In the introduction section, there is a subsection called "contributions " most of them are not original contributions to this work, the authors must be highlighted their contributions. 

3- In the related work section, the authors must summarize the previous related works and show what are the differences between the proposed work and previous works.

4- The Methods and Materials section, should rewrite in a more professional way. 

5- In the results section, there are many figures and tables without discussion and explanation. 

7- The conclusion section is too short, and must be revised. 

8- Overall, there are many subsections that may focus the readers, for that the authors must carefully address these issues in the revised version. 

Author Response

Kindly find the attached file

Round 2

Reviewer 2 Report

The author has addressed my comments, but still there one issue which is all the figures must be improved the current quality is not acceptable.

Author Response

Thank you for your valuable suggestion. We have updated the figures accordinlgy. However, there are some figures which we got from Python (experimental outputs), and we are unable to change its quality. Thanks